# Critical Review on the Progress of Plastic Bioupcycling Technology as a Potential Solution for Sustainable Plastic Waste Management

**DOI:** 10.3390/polym14224996

**Published:** 2022-11-18

**Authors:** Passanun Lomwongsopon, Cristiano Varrone

**Affiliations:** Section of Bioscience and Engineering, Department of Chemistry and Bioscience, Aalborg University, Fredrik Bajers Vej 7H, 9220 Aalborg, Denmark

**Keywords:** bioconversion, bioupcycling, fermentation, plastic upcycling, recycling, valorization

## Abstract

Plastic production worldwide has doubled in the last two decades and is expected to reach a four-fold increase by 2050. The durability of plastic makes them a perfect material for many applications, but it is also a key limitation to their end-of-life management. The current plastic lifecycle is far from circular, with only 13% being collected for recycling and 9% being successfully recycled, indicating the failure of current recycling technology. The remaining plastic waste streams are thus incinerated, landfilled, or worse, mismanaged, leading to them leaking into the environment. To promote plastic circularity, keeping material in the loop is a priority and represents a more sustainable solution. This can be achieved through the reuse of plastic items, or by using plastic waste as a resource for new materials, instead of discarding them as waste. As the discovery of plastic-degrading/utilizing microorganisms and enzymes has been extensively reported recently, the possibility of developing biological plastic upcycling processes is opening up. An increasing amount of studies have investigated the use of plastic as a carbon source for biotechnological processes to produce high-value compounds such as bioplastics, biochemicals, and biosurfactants. In the current review, the advancements in fossil-based plastic bio- and thermochemical upcycling technologies are presented and critically discussed. In particular, we highlight the developed (bio)depolymerization coupled with bioconversion/fermentation processes to obtain industrially valuable products. This review is expected to contribute to the future development and scale-up of effective plastic bioupcycling processes that can act as a drive to increase waste removal from the environment and valorize post-consumer plastic streams, thus accelerating the implementation of a circular (plastic) economy.

## 1. Introduction

Demand for plastic has dramatically increased during the last decades and continues growing, reaching 460 Mt in 2019, thus doubling from the 234 Mt reported in 2000 (Figure 1) [1]. Plastics are used in a wide variety of products, dominating and outperforming other materials, as they are versatile, cheap, lightweight, and resistant; however, they are also very diverse and typically designed for endurance rather than recyclability, which often makes the end-of-life management of these materials rather challenging. As a consequence, we’ve witnessed an increased accumulation of plastic wastes in the environment and the phenomenon has now reached such an extent so as to be recognized as a global problem.

To give some examples, according to the United States Environmental Protection Agency (US EPA), in 2018, 12.2% of municipal solid waste in the United States was made of plastics, the majority of which (75%) were landfilled (Figure 2). The statistics also showed that landfilled plastics have been increasing every year since 1960, while the recycling rates are still relatively low (a little higher than 8%) [2]. Somewhat better statistics are observed in the European Union (EU), where 35% is reported to be collected for recycling, though most plastic still goes to combustion for energy recovery (42%), while the rest (23%) is landfilled [3]. Unfortunately, only a fraction of these collected 35% is then really recycled at the highest level possible, with process losses and downcycling resulting in further reduction of the actual global recycling flow [4]. On a global scale, a disappointing 13% is actually collected for recycling [1].

The mismanagement of end-of-life plastics has thus become a threat to the environment and our health, with estimated 22 Mt of global plastic leakage into the environment in 2019 [1]. By 2050, it is expected that the production and incineration of plastic could release 2.8 gigatons of CO_2_ per year, equal to the emissions from 615,500-megawatt coal plants [5]. For the overall plastic pollution in the marine biosphere, more than 123 Mt of plastics leaked into the ocean from 1950–2015 (Figure 3), which is the reason for the death of 1 million sea birds and 100,000 sea animals yearly [6]. According to the study of Liebmann et al., PP, PET, PS and PE were detected in fecal samples of tested participants consuming seafood, implying that plastic pollution has already started to affect human health [7]. In fact, microplastics could potentially cause alterations in human chromosomes, leading to infertility, obesity and cancer [8].

Notably, the problems of the current plastic sector are not only related to the management of end-of-life plastic, but also the considerable use of virgin fossil resources in plastic production, which is accelerating the world to petroleum depletion. The linear flow of the plastics value chain (produce–use–discard) is among the main causes of the above-mentioned negative impacts. The European Commission has therefore recently introduced its circular economy action plan, to encourage a more sustainable value chain and ensure that post-consumer waste is kept in the loop (at the highest possible level) for as long as possible [11]. To reach this goal, new technological solutions are needed, with recycling strategies that help improving the techno-economic feasibility of the recycled plastics (still not competitive with virgin fossil ones [12] or that lead to improved value/properties of the new plastics (defined as “upcycling”) [13]. In fact, the currently implemented recycling technology mainly transforms the plastic waste into lower-value products (downcycling) or, at best, into the same level. In order to boost the development of these new technologies for plastic upcycling, the EC has been financing a significant amount of research and innovation projects, addressing both, chemical (i.e., iCAREPLAS, https://www.icareplast.eu (accessed on 7 March 2022); MultiCycle, http://multicycle-project.eu (accessed on 7 Mar 2022)) and biological processes (UPLIFT, https://upliftproject.eu (accessed on 7 March 2022); PRESERVE, https://www.preserve-h2020.eu (accessed on 7 March 2022); UpPE-T, https://uppet.eu (accessed on 7 March 2022); MIX-UP, https://www.mix-up.eu (accessed on 7 March 2022); BioICEP https://www.bioicep.eu (accessed on 7 March 2022); MIPLACE, https://miplacebio.com (accessed on 7 March 2022); Enzycle, https://www.enzycle.eu (accessed on 7 March 2022)).

Biological plastic upcycling is performed by using plastic waste streams as a carbon substrate for biotechnological processes, similarly to the approach used for lignocellulosic feedstocks [14]. Several enzymes have been identified that present hydrolytic properties to depolymerize certain plastics, while microbial fermentation processes can be developed to convert depolymerized plastics to higher-value products, for example, biopolymers [15,16,17]. Nevertheless, the biological route is generally hindered by plastic recalcitrance, typically related to hydrophobicity, and the crystallinity. The biodegradability of plastics can be very different, depending on their chemical structure: polyolefins such as polyethylene (PE) and polypropylene (PP) (which are among the most abundant plastics, Figure 4), for instance, cannot be easily degraded by microorganisms because of their recalcitrant C–C bonds, while polyesters are much more prone to be attacked at the hydrolyzable ester units. Several other factors, e.g., surface conditions, molecular weight, thermal properties, etc., also affect the biodepolymerization of plastic [18].

Most of the reviews dealing with plastic waste management focus on conventional processes, e.g., landfilling, incineration, mechanical, and chemical recycling [19,20,21] that typically downcycle the materials. They do not analyze the advantages of integrated processes that combine thermochemical and biochemical technologies. Up to date, relatively few studies have been focusing on plastic bioupcycling strategies; however, new low-cost biotechnological processes could act as fundamental drives for the valorization of plastic waste streams that currently are not effectively recycled. In fact, by converting such streams into higher-value upcycled polymers, the post-consumer plastic would no longer be regarded as a waste but as a valuable feedstock, thus contributing to keeping the materials in the loop. Therefore, the overall aim of this paper is to provide a comprehensive and critical review on plastic bioupcycling by identifying the potential strengths and limitations of these technologies, and thus speeding up the development of a more sustainable and economically feasible plastic sector. The review is structured based on different plastic types, where the depolymerization and fermentation steps are indicated separately, to highlight the challenges and technological solutions of the different steps.

## 2. Recycling of Conventional Plastic Wastes

Conventional recycling methods cannot keep up with the increasing amount and variety of plastic wastes. *Mechanical recycling* mainly involves grinding and pelletizing of relatively clean plastic streams into small particles, which are then reformed into new products without significant changes in the chemical structure. It represents the most mature recycling technology, with relatively low greenhouse gas emissions, and has been extensively reported in the literature [22]. Examples of mechanical recycling include the remolding of polyolefins waste for outdoor furniture, decking, and fencing, or of polyethylene terephthalate (PET) bottles to shoes. However, in the real situation, large streams of post-consumer plastic waste come in the form of (contaminated) mixed plastics, e.g., multilayer films (combination of different polymer types), galvanized polymers (combination of plastic with other materials such as metals, carbon, glass fibers, etc.), and additives (such as flame retardants and plasticizers) [23,24,25]. Mechanical recycling requires a sorting and cleaning process before recycling, which makes the handling of contaminated and/or mixed plastic streams extremely challenging, often leading to downcycling into less-valuable products. The number of cycles is also limited, due to the deterioration of plastic materials [26], and as a consequence, new technologies such as chemical recycling have been developed to reduce such limitation.

*Chemical recycling* includes thermochemical and catalytic conversions such as pyrolysis, gasification, fluid-catalyzed cracking, hydrocracking, and chemolysis (glycolysis, hydrolysis, methanolysis, aminolysis) [27]. These processes break down the polymer at high temperatures, with or without catalysts, to a mixture of oligomers/monomers and/or gaseous products and are thus applicable for handling heterogenous and contaminated plastic [28]. However, these recycling methods are usually costly, frequently use very large amounts of chemicals, and/or are energy-intensive, leaving behind hazardous gases and toxic residues [29]. Notably, chemical conversion emits more greenhouse gases (per ton of plastic utility treated) than most other treatment types, with the only exception of incineration.

Overall, even though conventional recycling methods reduce the amount of plastic going to landfills, the existing technologies often tend to reduce the properties of plastic waste-derived monomers and accelerate their end-of-lifetime [30,31]. As a consequence, new studies are starting to present *biotechnological recycling* strategies as a much-needed complementary solution to the end-of-life of those challenging plastic waste streams that currently are not effectively recycled. In fact, applying biological depolymerization and bio-recycling to plastics waste provides the opportunity to produce higher-value products in more sustainable and more mild conditions (lower temperature and energy requirements, absence of toxic chemicals, etc.), without the need of previous sorting. A good example is the new biotechnology developed by Carbios and the University of Toulouse that enables the efficient enzymatic depolymerization of post-consumer PET bottles on an industry-relevant scale and processing time (90% depolymerization into monomers in only 10 h) [32].

## 3. Bioupcycling

Besides biotechnological recycling, new upcycling processes are now extensively studied and under development. The possibility to convert post-consumer PET to polyhydroxyalkanoate (PHA) by enzymatic depolymerization and subsequent bacteria fermentation [16], for instance, allows researchers to obtain a bio-material with good technical substitution potential, novel properties (depending on the co-polymer) and biodegradability. From this perspective, post-consumer plastic can be upcycled rather than only recycled. Moreover, biological methods have the advantage that they can be applied to contaminated plastic waste (i.e., food or soil) and do not require previous separation of the different fractions. Furthermore, the high selectivity of enzymes could allow for a stepwise removal of specific components of the mixed-plastic waste, facilitating the downstream processing; thus, it can go beyond the limits of mechanical and chemical recycling [33].

Recent research in the biodegradation of plastic waste allowed for the establishment of plastic biodepolymerization processes for some plastic types, thus paving the path for greener plastic recycling processes [34,35,36,37,38,39,40,41].

Both natural and engineered enzymes for plastic depolymerization have been extensively studied [32,34,42,43,44,45,46]. Even though the cost of the enzymes is still of concern, a recent study on enzymatic recycling of PET has determined that it should only contribute to 4% of the overall operating costs [47]. The authors highlighted that an enzyme-based recycling process can be cost-competitive, and the constant development of enzyme performance and optimization of the process remains an opportunity to further improve the economic viability of this process. Once bioprocesses for bulk enzyme production have been established, the enzymatic degradation of plastic is a promising technology that will be implemented in the near future. Depolymerization enzymes can be utilized in multiple ways, including free enzymes, immobilized enzymes, extracellular enzymes of whole-cell biocatalysts, surface enzymes, and/or in the form of enzyme cocktails [48]. Moreover, synthetic biology can be applied to improve the catalytic activity of enzymes through protein engineering, e.g., direct evolution and rational protein design [49]. Plastic-degrading enzymes in microorganisms can evolve from their natural activity on recalcitrant biopolymers such as lignocellulose, chitin, and cutin. Polyester plastics could, for instance, be depolymerized by hydrolytic enzymes produced by bacteria or fungi, such as cutinases, esterases, lipases, ureases, and proteases, as they have hydrolyzable ester bonds in their backbone (Satti and Shah, 2020). An excellent example is the bacterial enzyme polyurethane hydrolases (PUase), capable of hydrolyzing polyurethane (PU) [14], or the well-known PETase and MHETase from *Ideonella sakaiensis* 201-F6, able to hydrolyze PET [45]. For plastics with a non-hydrolyzable C–C backbone, such as polyolefins (PE and PP), oxidative enzymes play a significant role in introducing active functional groups into the backbone, which consequently can undergo biodegradation [50]. Alkane hydroxylases from isolated *Pseudomonas* sp. E4 was also reported to play an important role in PE biodegradation [51]. This enzyme acts on the hydroxylation of C–C bonds to release primary or secondary alcohols, which are then further oxidized to ketones or aldehydes, and finally to carboxylic acids [52]. Thus, the identification and optimization of efficient enzymes represents a prerequisite for the development of bioupcycling processes. In fact, the enzymes are not going to use the polymers as a carbon source, as would happen with a microbial cell that degrades the polymers. This allows to fully recover the monomers for the subsequent upcycling step, for instance through the contribution of fermentation processes [13].

So, in conclusion, the established knowledge on biodepolymerization is expected to boost the development of new biocatalytic plastic upcycling processes, to produce value-added products and/or generate new (and more renewable) plastics with better properties, out of conventional plastic wastes. In this sense, the cost of renewable plastic production (which is still too high compared to conventional fossil-based plastics and related to feedstocks’ price and fluctuation) could be alleviated. Hence, renewable plastic will become more economically viable for general commodities, once large-scale production is reached. At present, bioupcycling with endogenous or engineered metabolic pathways has been demonstrated as a proof of concept and is a hot topic for researchers worldwide. The most relevant published studies have been reviewed in the sub-section below.

### 3.1. Bioupcycling of Polyethylene Terephthalate (PET)

PET is a petrochemical-based plastic produced on multimillion tons worldwide. It is a polyester made of the repeating units of ethylene glycol (EG) and terephthalic acid (TA). It is cheap and has a very low permeability to gas and moisture, making it an ideal material for single-use plastic bottles. The demand of PET worldwide is around 29 Mt in 2022 [53], while in Europe it reached 4.1 Mt in 2020 [3]. However, even if PET has a higher recycling rate than other plastic types (50% in Europe and 23% in the US for PET bottles), 69% of recycled PET was used for lower-grade applications such as trays, film, strapping, or fiber [54]. From these statistics, the PET management system clearly does not show a very high level of circularity; therefore, there is a clear need for new upcycling approaches to valorize PET into higher-value products or, at least, keep the material at the same level in the value chain.

Various upcycling strategies have been reported to valorize PET waste, including biotechnological processes and bioupcycling (Table 1). It typically combines depolymerization and fermentation/bioconversion strategies to produce new valuable products. Chemical processes (e.g., hydrolysis, alcoholysis, glycolysis, aminolysis, ammonolysis, and hydrogenolysis [55]) and thermal processes (e.g., hydrothermal liquefaction, pyrolysis, and microwave irradiation) are generally used for the depolymerization of PET. Interestingly, a considerable achievement through enzymatic or microbial degradation has also been reported, especially on PET, that can lead to the development of more sustainable bioupcycling processes. Since PET monomers are linked through hydrolyzable ester bonds, an increasing number of PET hydrolyzing (and/or surface modifying) enzymes have been reported recently. One of the major findings was, for instance, the discovery of *Ideonella sakaiensis* [45], a bacterium that can grow on PET as the only carbon source, thanks to the synergy of its enzymes *Is*PETase and MHETase that break down PET to mono-(2-hydroxyethyl)terephthalic acid (MHET), di-(2-hydroxyethyl)terephthalic acid (BHET), and finally to EG and TA. Subsequently, several studies reported the improvement of the mesophilic *Is*PETase through protein engineering, for example by increasing the hydrolytic activity [56] or the thermal stability, using rational protein engineering [34,43,57].

In order to boost enzyme production, several studies have been investigating the addition of signal peptides to the *Is*PETase gene to enhance its secretion during heterologous expression in *Escherichia coli* [58]. The use of heterogeneous immobilized biocatalysis through the use of different binding modules and linkers has also been extensively investigated [59,60]. Some of the most effective enzymes that act on PET hydrolysis are thermostable cutinases such as *Humicola insolens* cutinase (HiC) [35], *Thermobifida fusca* cutinase (TfCut2) [46], and leaf-branch compost cutinase (LCC) [61]. Improving their hydrolysis activity has also been reported using various methods, for example, by enhancing electrostatic interaction between TfCut2 and PET by cationic surfactant additive-based approach, and showed impressive biodegradation of PET by 97% within 30 h [36]. In general, the design of thermostable hydrolases has been intensively investigated during the last years [62] and resulted in the development of promising processes for enzymatic depolymerization. One of the biggest breakthroughs so far is probably represented by the study by Tournier and colleagues that engineered the LCC cutinase and increased its optimal reaction temperature up to the glass transition temperature (Tg) of PET [32]. Such bioprocesses can represent a valuable and sustainable alternative to thermochemical depolymerization and are laying the basis for the further conversion of plastic waste-derived monomers.

Bioupcycling post-consumer PET (or even polyolefins) to PHA has been gaining attention since this approach promotes the valorization of plastic waste into renewable biopolymers. PHA is a general term for microbial polyester of R-3-hydroxyalkanoic acids [63]. It is a promising substitute for several petroleum-based plastics, due to its superior thermal processability, biodegradability, and biocompatibility properties [64,65]. Kenny et al. reported using a two-step chemo-biotechnological process for upcycling PET to PHA [63]. First, the PET waste was hydrolyzed by pyrolysis at 450 °C at a feed rate of 1 kg/h, obtaining 77% of solid fraction (TA, oligomers, benzoic acid and others), 18% gas fraction (CO_2_, CO, H_2_, ethene, and others), and 6.3% liquid fraction (EG, acetic aldehyde and others). Second, the TA was dissolved in NaOH to generate sodium terephthalate, which was used as the sole carbon source to grow a PHA-accumulating strains. *Pseudomonas putida* GO16, *P. putida* GO19, and *P. frederiksbergensis* GO23 were found to consume sodium terephthalate and accumulated PHA up to 23–27% of CDW. GO19 was the most efficient at converting TA to PHA, with a productivity of 8.4 mgPHA/L/h. The research group continued to develop the bioprocess to enhance PHA production from TA by co-feeding with waste glycerol (WG) [66]. The fermentation was designed to have two distinct phases: the biomass growth phase and the PHA production phase. They found that when *P. putida* GO16 was fed with WG only during the growth phase (0–24 h) and WG and TA during the PHA production phase (24–48 h), the highest total PHA production was achieved (5.30 g/L). This strategy promoted a 2.0-fold higher PHA production than feeding with TA alone. This study showed that bioprocess engineering strategies are key to develop highly efficient bioupcycling of plastic waste.

Tiso et al. reported the bioupcycling of PET to PHA and hydroxyalkanoyloxyalkanoate (HAA) by using a combination of enzymatic hydrolysis and whole-cell biocatalyst [16]. PET was hydrolyzed by LCC, a polyester hydrolase capable of efficient PET depolymerization to TA and EG. *Pseudomonas umsongensis* GO16 KS3 was found to consume both TA and EG (the latter at a 3.5-fold lower rate than TA) within 23 h of cultivation, in a 5 L bioreactor producing PHA. The PHA production only reached 0.15 g/L, or 7% of CDW. Interestingly, HAA was produced from TA only, while EG was exclusively used for growth. A maximum HAA concentration of 35 mg/L was achieved with a production rate and yield of 5 mg/L/h and 0.01 gHAA/gTA, respectively [16]. Despite the low performance, this represents a highly interesting approach with potential industrial applications (once optimized). In fact, HAA can be directly polymerized with 4,4′-methyl diphenyl diisocyanate and butanediol (BDO) to form biopoly(amide urethane) (bio-PU).

Another study reported the bioconversion of TA, a monomer of PET, to muconic acid (MA), which is currently used to produce adipic acid (AA), an important monomer for various plastics [67]. PET was first depolymerized by microwave radiation at 230 °C for 50 min to TA and EG. Then, the *E. coli* strain CTL-1 (expressing TphAabc, TphB, and AroY, which is responsible for converting TA to catechol, via 1,2-dihydroxy-3,5-cyclohexadiene-1,4-dicarboxylate (DCD) and protocatechuic acid (PCA)) and the strain MA-1 (expressing CatA, which is a catechol 1,2-dioxygenase responsible for converting catechol to MA), were combined for MA synthesis. The MA concentration reported was 2.7 mM, accounting for a 85.4% molar yield (MA/TA). In the same study, gallic acid (GA), pyrogallol, vanillic acid (VA), and glycolic acid (GLA) were also produced from engineered stains, using TA. In addition, GLA was produced by EG-fermenting *Gluconobacter oxydans* KCCM 40109.

The recent study by Sadler and Wallace (2021) showed the development of a one-pot bioprocess to convert TA from PET waste into a value-added molecule, vanillin. PET from a post-consumer plastic bottle was firstly hydrolyzed to TA by semi-purified LCC at 72 °C. Then, the reaction was cooled down and the engineered strain, *E. coli* RARE_pVanX, was added to perform the bioconversion. *E. coli* RARE_pVanX was constructed with two plasmids that were encoded for different enzymes (terephthalate 1,2-dioxygenase, dihydroxy-3,5-cyclohexadiene-1,4-dicarboxylic acid dehydrogenase, carboxylic acid reductase, and catechol O-methyltransferase), which convert TA to vanillin via intermediates (PCA, VA, and dihydroxybenzaldehyde). The process optimization was performed by screening the protein-expression media (M9-glucose supplemented with L-Met and n-butanol (nBuOH)), increasing *E. coli* cell membrane permeability to TA (addition of 1% *v*/*v* n-BuOH), adjusting pH (5.5) and temperature (22 °C), and mitigating the toxicity of vanillin by in situ product removal, using oleyl alcohol. The final production of vanillin reached 789 μM (119 mg/L) or 79% conversion. [68].

The chemo-biological upcycling of PET to the multifunctional coating material, catechol, was also reported. First, PET waste was glycolyzed to a mixture of PET oligomers. Then, an enzymatic hydrolysis of the glycolyzed products was performed, turning the mixture without previous purification into TA, by *Bacillus subtilis* esterase (Bs2Est). The catechol production from TA was consequently conducted, using a catechol biosynthesis strain (obtained through the combination of the TA degradation module and catechol biosynthesis module in *E. coli*). The final titer of catechol was 5.97 mM, accounting for 99.5% conversion by mol of TA. Catechol shows great functions as a coating material without the need for an adherence layer, and its antibacterial activity is comparable to chitosan [69].

**Table 1 polymers-14-04996-t001:** PET bioupcycling.

Depolymerization Strategy	Depolymerization Products Used as a Feedstock for Fermentation Step	Fermentation Strategy	Products from Fermentation	Titer	Productivity	Yield	Ref.
Hydrolytic pyrolysis at 450 °C	Solid product mixture (terephthalic acid (TA), oligomers, benzoic acid, and others)	Fermentation in shake flask containing 4.2 g/L of PET-derived sodium terephthalate and 67 mg/L of nitrogen at 30 °C for 48 h by *Pseudomonas putida* GO16	medium chain length PHA (mclPHA)	0.25 g/L	8.4 mgPHA/L/h	0.27 gPHA/gCDW	[63]
Hydrolytic pyrolysis at 450 °C	Solid product mixture (TA, oligomers, benzoic acid, and others)	Fermentation in shake flask containing 4.2 g/L of PET-derived sodium terephthalate and 67 mg/L of nitrogen at 30 °C for 48 h by *P. putida* GO19	mclPHA	0.25 g/L	8.4 mgPHA/L/h	0.23 gPHA/gCDW	[63]
Hydrolytic pyrolysis at 450 °C	Solid product mixture (TA, oligomers, benzoic acid, and others)	Fermentation in shake flask containing 4.2 g/L of PET-derived sodium terephthalate and 67 mg/L of nitrogen at 30 °C for 48 h by *P. putida* GO23	mclPHA	0.27 g/L	4.4 mgPHA/L/h	0.24 gPHA/gCDW	[63]
Pyrolysis	TA	Fed-batch fermentation in 19.5 L-stirred tank reactor with controlled pH of 6.9 and dissolved oxygen (DO) level above 40% at 30 °C for 48 h by *P. putida* GO16 supplied with TA as the sole growth and PHA substrate	mclPHA	2.61 g/L	0.05 g/L/h	0.30 gPHA/gCDW	[66]
Pyrolysis	TA	Fed-batch fermentation in 19.5 L-stirred tank reactor with controlled pH of 6.9 and DO level above 40% at 30 °C for 48 h by *P. putida* GO16 supplied with waste glycerol (WG) as growth substrate and TA as PHA substrate	mclPHA	5.22 g/L	0.11 g/L/h	0.36 gPHA/gCDW	[66]
Pyrolysis	TA	Fed-batch fermentation in 19.5 L-stirred tank reactor with controlled pH of 6.9 and DO level above 40% at 30 °C for 48 h by *P. putida* GO16 supplied with TA as the sole growth and PHA substrate	mclPHA	5.30 g/L	0.11 g/L/h	0.35 gPHA/gCDW	[66]
Pyrolysis	TA	Fed-batch fermentation in 19.5 L-stirred tank reactor with controlled pH of 6.9 and DO level above 40% at 30 °C for 48 h by *P. putida* GO16 supplied with WG as growth and PHA substrate and TA as PHA substrate only	mclPHA	4.98 g/L	0.10 g/L/h	0.35 gPHA/gCDW	[66]
Pyrolysis	TA	Fed-batch fermentation in 19.5 L-stirred tank reactor with controlled pH of 6.9 and DO level above 40% at 30 °C for 48 h by *P. putida* GO16 supplied with WG and TA as both growth and PHA substrates	mclPHA	4.42 g/L	0.09 g/L/h	0.36 gPHA/gCDW	[66]
Enzymatic degradation by recombinant leaf-branch compost cutinase (LCC)	TA, ethylene glycol (EG), mono-(2-hydroxyethyl)terephthalic acid (MHET), di-(2-hydroxyethyl)terephthalic acid (BHET)	Fermentation in 5 L-stirred tank reactor with controlled pH of 7.0 and DO level above 20% at 30 °C for 28 h *Pseudomonas umsongensis* GO16 KS3 supplied with hydrolyzed PET at the amount to yield 40 mM of TA and EG and limited inorganic nutrient	mclPHA	0.15 g/L	NA	0.014 gPHA/gSubstrate	[16]
Enzymatic degradation by recombinant LCC	TA, EG, MHET, BHET	Fermentation in shake flask containing Delf medium with diluted (1:20) hydrolyzed PET (TA and EG concentration of 15–18 mM) at 30 °C for 24 h by *P. umsongensis* GO16 KS3 pSB01	Hydroxyalkanoyloxy-alkanoate (HAA)	35 mg/L	5 mg/L/h	0.01 gHAA/gTA	[16]
Microwave radiation for 50 min at 230 °C	TA	Bioconversion by metabolically engineered *E. coli* strain PCA-1 and HBH-2 to convert TA to intermediate protocatechuic acid (PCA), and then to gallic acid (GA), at 30 °C and 250 rpm for 24 h in 50 mM Tris buffer (pH 7.0) containing 2% (*w*/*v*) glycerol	GA	2.7 mM	NA	0.925 M_GA_/M_TA_	[67]
Microwave radiation for 50 min at 230 °C	TA	Bioconversion by metabolically engineered *E. coli* strain PG-1a to convert TA to intermediate PCA, GA, and then pyrogallol, at 30 °C and 250 rpm for 6 h in 50 mM Tris buffer (pH 7.0) containing 2% (*w*/*v*) glycerol	Pyrogallol	1.1 mM	NA	0.327 M_Pyrogallol_/M_TA_	[67]
Microwave radiation for 50 min at 230 °C	TA	Bioconversion for 6 h by metabolically engineered *E. coli* strain CTL-1 and MA-1 to convert TA to intermediate catechol, and then to muconic acid (MA)	MA	2.7 mM	NA	0.854 M_MA_/M_TA_	[67]
Microwave radiation for 50 min at 230 °C	TA	Bioconversion using double-catalyst VA-2a system for 48 h by metabolically engineered *E. coli* strain PCA-1 and OMT-2^His^ to convert TA to intermediate PCA and then to vanillic acid (VA), in 50 mM Tris buffer (pH 7.0) containing 10% (*w*/*v*) glycerol, 10 g/L yeast extract, 20 g/L peptone, and 2.5 mM L-methionine	VA	1.4 mM	NA	0.416 M_VA_/M_TA_	[67]
Microwave radiation for 50 min at 230 °C	EG	Bioconversion by *Gluconobacter oxydans* KCCM 40109 using 10.7 mM of EG from PET hydrolysate as a feedstock, at 30 °C and 220 rpm in shake flask at the working volume of 1 L	Glycolic acid (GLA)	NA	NA	0.986 M_GLA_/M_EG_	[67]
-	EG (mock substrate to study upcycling of PET-derived monomer)	Fermentation in shake flask containing 10% (*v*/*v*) EG in 250 mM potassium phosphate buffer (pH 7.0) at 30 °C with gentle stirring and aeration at 1 VVM for 120 h by *Pichia naganishii* AKU 4267	GLA	105 g/L	NA	0.880 M_GLA_/M_EG_	[70]
-	EG (mock substrate to study upcycling of PET-derived monomer)	Fermentation in shake flask containing 10% (*v*/*v*) EG in 250 mM potassium phosphate buffer (pH 7.0) at 30 °C with gentle stirring and aeration at 1 VVM for 120 h by *Rhodotorula* sp. 3Pr-126	GLA	110 g/L	NA	0.922 M_GLA_/M_EG_	[70]
-	EG (mock substrate to study upcycling of PET-derived monomer)	Fermentation in shake flask containing 100 mM of EG in nitrogen limiting M9 medium (0.132 g/L of (NH_4_)_2_SO_4_) at 30 °C for more than 72 h by *P. putida* MFL185 (engineered strain that has the *tac* promoter inserted before the native glycolate oxidase operon and harbor overexpression)	mclPHA	NA	NA	0.32 gPHA/gCDW and 0.06 gPHA/gEG	[71]
-	EG (mock substrate to study upcycling of PET-derived monomer)	Anaerobic fermentation of 50 mM EG at 30 °C by acetogenic bacterium *Acetobacterium woodii*	Acetate	10.4 mM	3.6 μmol/mg/h	NA	[72]
-	EG (mock substrate to study upcycling of PET-derived monomer)	Anaerobic fermentation of 50 mM EG at 30 °C by acetogenic bacterium *A. woodii*	Ethanol	12.0 mM	4.8 μmol/mg/h	NA	[72]
Enzymatic degradation by semi-purified LCC (pH 10.0) at 72 °C for 48 h	PET hydrolysate	Bioconversion using metabolically engineered *E. coli* RARE_pVanX to convert TA to intermediate protocatechuate (PC), and then to vanillin using optimized condition: M9-glucose supplemented with L-Met and nBuOH as a protein expression media, pH 5.5, room temperature for 24 h, in situ product removal (ISPR) by oleyl alcohol	Vanillin	300–400 μM	NA	NA	[68]
-	TA (mock substrate to study upcycling of PET-derived monomer)	Bioconversion using metabolically engineered *E. coli* RARE_pVanX to convert TA to intermediate PC and then to vanillin using optimized condition: M9-glucose supplemented with L-Met and nBuOH as a protein expression media, pH 5.5, room temperature for 24 h, ISPR by oleyl alcohol	Vanillin	789 μM	NA	0.79 M_vanillin_/M_TA_	[68]
Chemical glycolysis at 200 °C for 3 h	Mixture of BHET, MHET, and PET oligomers at 84.8, 7.7, and 8.7%, respectively	Enzymatic hydrolysis of the glycolyzed products (the mixture) into TA by *Bacillus subtilis* esterase (Bs2Est) (2 U/mL at 30 °C and 1000 rpm), following by producing catechol from PET hydrolysates using a catechol biosynthesis strain that was established using the combination of the TA degradation module and catechol biosynthesis module in *E. coli* (in 12 h)	Catechol	5.97 mM	NA	0.995 M_Catechol_/M_TA_	[69]
Chemocatalytic glycolysis: PET was glycolyzed with EG as a solvent (1:4 *w*/*w*) and catalyzed by 1% (*w*/*w*) titanium (IV) butoxide at 220 °C overnight	BHET	Fermentation in 3 L-bioreactor with batch culture in the first 4 h fed with 4-hydroxybenzoic acid to induce the β-ketoadipate pathway, followed by fed-batch culture using BHET as a carbon source (pulse adding at 9.1, 23.3, 32.8, 48.2 and 73.9 h). The sequential metabolic engineered *Pseudomonas putida* KT2440 (constitutive expression of native genes for EG utilization, expression of gene for TA catabolism, expression of PETase and MHETase for BHET hydrolysis, and gene deletion to enhance β-ketoadipic acid production) was used for bioconversion.	β-ketoadipic acid	15.1 g/L	0.16 g/L/h	0.76 M_β-ketoadipic acid_/M_BHET_	[73]
Chemocatalytic glycolysis and enzymatic hydrolysis: PET was glycolyzed with EG as a solvent and catalyzed by betaine at 190 °C for 30–120 min, followed by enzymatic hydrolysis (PETase and MHETase)	TA	Whole cell bioconversion of TA (4.5 g/L) to protocatechuic acid by metabolically engineered *E. coli* PCA-1 was performed in shake flask at 30 °C and 250 rpm	PCA	3.8 g/L	-	0.904 M_PCA_/M_TA_	[74]
Chemocatalytic glycolysis and enzymatic hydrolysis: PET was glycolyzed with EG as a solvent and catalyzed by betaine at 190 °C for 30–120 min, followed by enzymatic hydrolysis (PETase and MHETase)	EG	Whole cell bioconversion of EG (30.6 g/L) to GLA by Gluconobacter oxydan KCCM 40109 was performed in shake flask at 30 °C and 200 rpm.	GLA	31.4 g/L	-	0.916 M_GLA_/M_EG_	[74]

### 3.2. Bioupcycling of Polyurethanes (PU)

PU is a general term used for a class of polymers typically derived from the polycondensation of (poly)isocyanates (−NCO) and polyols (exothermic reactions) [75]. There are three main types of PU: polyester, polycaprolactone, and polyether urethanes, depending on the type of long-chain diol used in PU production [76]. PUs are used in many fields of application, such as, building insulation, pillows and mattresses, and insulating foams for fridges [3]. In 2020, PU production was 3.81 Mt in Europe, representing 7.8% of total plastic production [3]. Since many PU types have a thermoset nature with covalent cross links, recycling is still extremely challenging. Almost 50% of PU produced in the European Union goes to a landfill, 45% being incinerated with energy recovery, and only 5% is mechanically recovered [77]. The statistics from the US show that the mechanical recycling of PU on a large scale has only been performed on flexible PU foams by shredding and mixing the scrap polymers with binders to produce padding-type products; such processes have the limitation that they can be repeated only 8–10 times, until reaching the end-of-life [78]; this represents a common problem with mechanical recycling. Chemical recycling methods used for PU may include hydrolysis, glycolysis, and aminolysis, which use water, glycol, and ammonia or ammonium hydroxide as a nucleophile, respectively [79]. Europe’s first chemical recycling plant for post-consumer PU foam was constructed very recently in Semoy, France, by the Dow Chemical Company and partners, employing glycolysis for recovering polyols [80]. Another method called acidolysis, based on a combination of two carboxylic acids and an unspecified non-metallic catalyst, has been successfully employed by H&S, a Poland-based company, to break up PU and recover polyols [81]. Even though these processes can recover chemical constituents from PU waste, the negative impact from the toxic isocyanides and the huge amount of chemical catalysts consumption cannot be neglected. Therefore, despite the recalcitrance of these polymers, biodegradation of PU using microorganisms (i.e., fungal species) or enzymes is gaining more interest lately, as a more environmentally friendly and sustainable alternative [82,83]. On the other hand, since PUs represent a large class of plastics, it is difficult to point to a single enzyme that can degrade them all. In the presence of hydrolase-type enzymes, e.g., lipase, esterase, and protease, PUs can be hydrolyzed by a three-step mechanism: (1) chemical dissolution of ester and amide bonds in the polymer chain; (2) decreasing molecular weight and viscosity; (3) cleaving all polymer chains [76]. However, urethane forms stable bonds that can be hydrolyzed at a slower rate than ester bonds [84]. *Bacillus subtilis* strain MZA-75 [85] and *Pseudomonas aeruginosa* strain MZA-85 [86] were reported for polyester-based PU degradation. These two strains were also studied in bacterial co-culture and found to utilize polyester PU as a carbon and energy source more efficiently, as compared to the individual strains. The main degradation products were BDO and AA [87]. Another study investigated PU degradation by the enriched mixed-microbial consortia and reported the presence of extracellular enzymes, such as esterases, proteases, and ureases, affecting ester, urethane, ether, aromatic, and aliphatic groups of the plastic [88]. Fungi, such as *Aspergillus tubingensis* [89], *Penicillium* sp. [90], *Cladosporium cladosporioides* complex [91], or fungal communities [92] were also reported for their ability to degrade PU.

Besides BDO and AA, other PU degradation products are EG, 2,4′-toluenediamine (TDA), and 4,4′-methylenedianiline (MDA). They can be used to synthesize new PU or other polyesters, e.g., PHA, polybutylene succinate (PBS), poly(1,3-propylene succinate-*ran*-1,4-butylene succinate) (PPBS), and poly(1,3-propylene adipate-*ran*-1,4-butylene adipate) (PPBA) [79].

Last but not least, a recent study reported the upcycling processes of PU waste to rhamnolipid biosurfactants, [75] (Table 2). They successfully showed the growth of a defined mixed culture composed of three *Pseudomonas putida* KT2440 strains obtained by adaptive laboratory evolution on mock PU hydrolysate. The highest specific growth rate was 0.21 h^−1^ on the mixture of AA, 1,4-BDO, and EG. After engineering of the mixed culture, 0.10 g/L of rhamnolipids were produced from the AA, BDO, and EG mixture.

### 3.3. Bioupcycling of Polyolefins

Polyolefins represent the most abundant polymers, accounting for 42% of all plastics globally produced and 50% of plastics produced in Europe (Figure 4). In the packaging sector, around 70% of the total plastic waste generated is made of polyolefins [1]. The highest demand of polyolefins is PE (LLDPE, LDPE, MDPE, and HDPE), followed by PP. Polyolefins are non-polar materials, durable, chemically resistant, and have low permeability; thus, they are used in various applications, especially for flexible food packaging [94]. However, their durability also make them retain in nature for a very long time (the half-life can be up to 5000 years on land) [95]. This is due to the stable carbon to carbon bonds of the polyolefins that make biodegradation much more challenging than in polyesters (e.g., PET) [96]. However, due to its abundance in our waste, it is imperative to look into bioupcycling strategies for this type of plastic. Instead of mineralization or depolymerization to gaseous monomers, polyolefins’ recalcitrancy could be seen as an opportunity for upcycling by recovering and converting oligomers released during biofragmentation.

#### 3.3.1. Polyethylene (PE)

A major type of polyolefins used globally is PE, accounting for around 25% of the market share [1]. It is widely used in various applications such as plastic bags, composite films, food and drink packaging, and bubble wraps. PE is a very high-molecular-weight hydrocarbon plastic with recalcitrant C–C bonds; thus, biodegradation of PE needs to be initiated by a pre-oxidation step, i.e., physicochemical oxidation with the formation of free radicals. The radicals can react with oxygen and form peroxyl groups under aerobic conditions, while they create end terminal unsaturated double bonds under anaerobic conditions [95]. The hydroperoxyl group is highly reactive and can generate several types of oxygen-containing products, including carbonyl groups. The oxidation of polyolefins can be followed by FTIR through the carbonyl peak formation and expressed as a carbonyl index (the ratio of carbonyl absorbance (1715 cm^−1^) to an invariant absorbance characteristic) [97].

The polymer containing oxygenated groups is much more prone to undergo biodegradation; however, not one but a cascade of enzymes is thought to be involved in polyolefins biodegradation. Studies reported that oxidative enzymes such as peroxidases, oxygenases, and laccases could play a significant role in PE biodegradation [98,99,100,101,102,103,104]. A laccase from *Rhodococcus ruber* C208, for instance, was found to reduce 20% and 15% of the average molecular weight (Mw) and average molecular number (Mn) of PE and increase the carbonyl group formation in PE structure, as evidenced by gel permeation chromatography (GPC) and attenuated total reflection Fourier-transform infrared spectroscopy (ATR-FTIR) [104]. A recent study by Yao et al. reported different effects of laccases from *Botrytis aclada* (BaLac) and *Bacillus subtilis* (BsLac) on high-temperature UV-irradaited PE, depending on their different redox potential. New functional groups, including -OH, -C=O, and C=C, were detected in PE chains after exposure to the laccase-mediator system. The Mw of PE was reduced by 40% with the BaLac degradation. The degradation products were aldehydes, ketones, and alcohols between C4–C8 (BaLac-treated) and C7–C15 (BsLac-treated) [37].

Alkane hydroxylases (AHs) were also reported to be involved in polyolefins degradation. AHs are known to convert terminal CH_3_ of alkane to 1-alkanol, which then can be oxidized by several other enzymes to fatty acids and then enter the β-oxidation pathway to provide energy for microbial growth [105]. AH systems include alkane monooxygenase (AlkB), electron-transport protein rubredoxin, and rubredoxin reductase, catalyzing hydroxylation of alkanes at the terminal carbon (ω-position) in aerobic bacteria [40,106]. Jeon and Kim showed that AHs from *P. aeruginosa* E7 are involved in the biodegradation of low-molecular-weight PE, confirmed by the conversion of approximately 19% of PE into CO_2_ during biodegradation at 37 °C, using an engineered *E. coli* strain expressing AH [40]. They also investigated the effect of induction conditions: the addition of *n-*hexadecane induced the transcription level of alkB1 in *E. coli*, whereas alkB2 was induced by both *n-*hexadecane and *n-*dodecane [107]. Microbial oxidation of PE is proposed to be similar with the metabolic pathway of linear *n-*alkanes, as shown in three pathways below [108,109].

^1.^ Terminal oxidation:

RCH_3_ → RCH_2_OH → RCHO → RCOOH

^2.^ Bi-terminal oxidation:

H_3_CRCH_3_ → H_3_CRCH_2_OH → H_3_CRCHO → H_3_CRCOOH → HOH_2_CRCOOH → OHCRCOOH → HOOCRCOOH

^3.^ Subterminal oxidation:

RCH_2_CH_2_CH_3_ → RCH_2_CH(OH)CH_3_ → RCH_2_C(O)CH_3_ → RCH_2_OC(O)CH_3_ → RCH_2_OH + CH_3_COOH

Cytochromes P450, an iron (Fe)-containing heme protein, among the most versatile biocatalysts in nature, are as also foreseen to be part of the PE-degrading machinery [106]. They are monooxygenases that are able to introduce one atom of O_2_ into a wide variety of organic substrate molecules. P450s can catalyze many reaction types: C-H hydroxylation; C=C double bond epoxidation; heteroatom oxygenation; O-, N-, and S-dealkylation; aromatic coupling; and C-C bond cleavage [106]. In particular, they can hydroxylate linear alkanes, alcohols, and fatty acids of various chain lengths [110], which are intermediates in the proposed degradation pathway of polyolefins. However, further effort is still needed to prove the exact mechanism of these enzymes. In addition to the above-mentioned enzymes, intracellular enzymes including alcohol dehydrogenase, aldehyde dehydrogenase, Baeyer–Villiger monooxygenase, and esterase were also reported to be involved in polyolefins biodegradation to alcohol, aldehydes, ketones, and aliphatic acids, which can then be further metabolized by bacteria via the β-oxidation pathway and subsequently enter the citric acid cycle [52,111].

Recently, a few publications have reported new innovative bioprocesses for PE biodegradation. Peixoto et al. (2022) reported new evidence on microbial oxidation of PE by nitric oxide (NO) produced from nitrifier and denitrifying bacteria. Three genera, including *Comamonas* sp., *Delftia* sp., and *Stenotrophomonas* sp. can oxidize ammonia (NH_4_) to nitrite (NO_2_), releasing NO as an intermediate, which oxidizes PE through the introduction of a nitro group into the PE chain (the high peak at 1550 cm^−1^ detected by FTIR) [112]. Furthermore, a latex clearing protein (Lcp_K30_) from *Streptomyces* sp. strain K30 was reported to fragment UV-pretreated PE and PP, by adding -OH and -C=O functional groups to the polyolefins’ backbone. The Mw of PE was reduced by 42%, and ketones were detected by GC-MS as degradation products [38].

Several other microorganisms have been identified during PE biodegradation, including, for example, bacteria (*Bacillus*, *Pseudomonas*, *Ralstonia*, *Rhodococcus*, *Staphylococcus*, *Streptomyces* and *Stenotrophomonas*), fungi (*Aspergillus*, *Cladosporium*, *Lentinus*, *Phanerochaete* and *Penicillium*), and gut microbiomes of mealworms and waxworms [113,114,115,116,117,118]. The very recent publication by Sanluis-Verdes and team reported the first discovery of animal enzymes, belonging to the phenol oxidase family, obtained from saliva of waxworms’ larvae (*Galleria mellonella*) that are capable of oxidizing and depolymerizing PE. The authors reported an increase in the carbonyl index and changes in the average molecular weight of PE, together with the release of oxidized aliphatic ketones as degradation products after saliva contact [39]. A process utilizing different microorganisms and enzymes was recently patented by the Technical Research Centre of Finland VTT, that included (among other things) the use of *Bacillus licheniformis*, *Bacillus flexus*, *Bacillus subtilis,* and *Bacillus cereus*’ metal-dependent hydrolases [119]. Nevertheless, in contrast with polymers made of ester bond linkages, major advancements in enzyme discovery and optimization are still needed to reach a breakthrough in the case of polyolefins.

Consequently, as biodegradation/depolymerization of PE is still being developed, upcycling of PE has been so far mainly reported as a combination of thermochemical and biological process. Different studies have tried to develop processes for bioupcycling PE waste into biodegradable bioplastic PHA (Table 3). Guzik and colleagues, for instance, examined a two-step chemo-biotechnology conversion. Firstly, PE was pyrolyzed to a mixture of hydrocarbons (C8–C32), called PE hydrolysis wax, which is composed of 90% alkanes and 10% alkenes. The PE pyrolysis wax was then used as a sole carbon source to grow PHA-accumulating strains. Next, the PHA production was enhanced by changing the nitrogen source from NH_4_Cl to NH_4_NO_3_ and adding a biosurfactant (rhamnolipid). The authors discussed that rhamnolipids enhanced the uptake of aliphatic hydrocarbon. The highest PHA was produced from *Pseudomonas aeruginosa* GL-1 with 18.9% of CDW (0.074 g/L) [120].

Another study performed oxidative degradation of PE at 145 °C to obtain an oxidized polyethylene wax (O-PEW), which was used as a novel carbon source. *Ralstonia eutropha* H16 was cultured in tryptone soya broth (TSB) with 4 g/L O-PEW to produce PHA. The results showed 1.24 g/L PHA production after 48 h, accounting for 33.8% CDW [121]. Researchers also investigated the use of non-oxidized PE wax (N-PEW), which is cheaper and easier to produce than O-PEW [122]. Not surprisingly, a lower PHA production was found, with 0.46 g/L (32% CDW). The presence of less recalcitrant carbon sources in O-PEW, such as fatty and carboxylic acids, could be the reason that makes it more accessible to bacteria. There are some studies investigating the direct use of untreated PE to produce PHA by different strains, such as *Cuprividus necator* H16, *Pseudomonas putida* LS46, and *Acinetobacter pittii* IRN19; however, the reported PHA yields are very low [109]. To date, depolymerization of PE by thermal processes seems to be one of the most effective strategies to allow for efficient (subsequent) microbial usage. To open the window for efficient polyolefins bioupcycling technologies, the development of cheap pretreatments and more efficient biodegradation processes are still needed.

**Table 3 polymers-14-04996-t003:** PE bioupcycling.

Depolymerization Strategy	Depolymerization Products Used as a Feedstock for Fermentation Step	Fermentation Strategy	Products from Fermentation	Titer	Productivity	Yield	Ref.
Pyrolysis	PE hydrolysis wax (a mixture of hydrocarbons (C8–C32): 90% alkanes and 10% alkenes)	Fermentation in shake flask containing 0.05% (*w*/*v*) PE pyrolysis wax as a sole carbon source and 0.025% (*w*/*v*) of NH_4_Cl as a nitrogen source at 30 °C for 48 h by *Pseudomonas aeruginosa* GL-1	PHA	0.023 g/L	NA	0.10 gPHA/gCDW	[120]
Pyrolysis	PE hydrolysis wax (a mixture of hydrocarbons (C8–C32): 90% alkanes and 10% alkenes)	Fermentation in shake flask containing 2% (*w*/*v*) PE pyrolysis wax as a sole carbon source and 0.019% (*w*/*v*) of NH_4_NO_3_ as a nitrogen source at 30 °C for 48 h by *P. aeruginosa* GL-1, in the presence of 0.05% (*w*/*v*) rhamnolipids	PHA	0.074 g/L	NA	0.19 gPHA/gCDW	[120]
Pyrolysis	PE hydrolysis wax (a mixture of hydrocarbons (C8–C32): 90% alkanes and 10% alkenes)	Fermentation in shake flask containing 2% (*w*/*v*) PE pyrolysis wax as a sole carbon source and 0.019% (*w*/*v*) of NH_4_NO_3_ as a nitrogen source at 30 °C for 48 h by *P. aeruginosa* PAO1, in the presence of 0.05% (*w*/*v*) rhamnolipids	PHA	0.045 g/L	NA	0.15 gPHA/gCDW	[120]
Oxidative degradation in a two-phase system (gas-liquid phase), after melting at 145 °C and using oxygen	Oxidized polyethylene wax (O-PEW)	Fermentation in shake flask containing 4 g/L melted O-PEW emulsified in TSB by sonication as a sole carbon source at 30 °C for 48 h by *Ralstonia eutropha* H16	PHA	1.25 g/L	NA	0.34 gPHA/gCDW	[121]
-	-	Fermentation in shake flask containing Ramsey’s media with 1% LDPE particles at 30 °C and 150 rpm for 21 d by *Cuprividus necator* H16	short chain length PHA (sclPHA)	NA	NA	0.0318 gPHA/gCDW	[123]
-	-	Fermentation in shake flask containing Ramsey’s media with 1% LDPE particles at 30 °C and 150 rpm for 21 d by *Pseudomonas putida* LS46	mclPHA	NA	NA	0.0054 gPHA/gCDW	[123]
-	-	Fermentation in shake flask containing Ramsey’s media with 1% LDPE particles at 30 °C and 150 rpm for 21 d by *Acinetobacter pittii* IRN19	mclPHA	NA	NA	0.0049 gPHA/gCDW	[123]
Pyrolysis	Non-oxidized PE wax (N-PEW)	Fermentation in shake flask containing 4 g/L melted N-PEW emulsified in TSB by sonication as a sole carbon source at 30 °C for 48 h by *Cupriavidus necator* H16	PHA	0.46 g/L	NA	0.32 gPHA/gCDW	[122]

#### 3.3.2. Polypropylene (PP)

PP is the second most produced plastic after PE, contributing to 16.6% of global plastic [1]. It is used for various applications, mainly food packaging, automobile, and the textile industry. The large production of PP, its recalcitrance, and the short application lifespan creates a large amount of post-consumption waste, posing a threat to the environment.

PP is categorized as a polyolefin, along with PE, that has hydrophobic backbones composed of long carbon chains and high molecular weight [52]. The enzyme machinery required to degrade PP is similar to PE; however, the presence of a methyl side-chain in PP makes it even more resistant to microbial or enzymatic depolymerization [124]. Similar to other polyolefins, increasing PP hydrophilicity by oxidation should, in principle, enable microbial colonization on its surface and initiate degradation through the extracellular enzymes.

PP biodegradation has been reported on a range of screened microorganisms. Jeon and Kim isolated a PP degrading strain from the municipal solid waste stack field and identified it as *Stenotrophomonas panacihumi* PA3-2. The strain degraded 20.3% and 12.7% of low- and high-molecular-weight PP, respectively, at 37 °C and a period of 90 days [124]. The single strain *Enterobacter* sp nov. bt DSCE01, *Enterobacter cloacae* nov. bt DSCE02, and *Pseudomonas aeruginosa* nov. bt DSCE-CD03 isolated from cow dung also showed similar degradation of PP and LDPE (less than 20%) in the study of Skariyachan et al. [125]. However, when they were grown together as a consortium, 63% and 64% degradation of PP and LDPE were found, respectively. A similar result was reported by using thermophilic consortia of *Aneurinibacillus aneurinilyticus*, *Brevibacillus agri*, *Brevibacillus* sp. and *Brevibacillus brevis,* which degraded 56% of PP [126]. Further PP biodegradation advances were reported using endophytic fungi [99] and gut microbiota of invertebrate larvae [127]. So far, a superior biodegradation efficiency was obtained by using mixed culture. Skariyachan et al. reported the formation of aldehyde and methyl groups, as well as of *cis*-2-chlorovinyl acetate, tri-decanoic acid, and octa-decanoic acid products from polyolefins biodegradation [126]. Such observations are aligned with the proposed metabolism of polyolefins, which is expected to start with an oxidation that leads to the formation of carbonyl groups, followed by a further oxidation to aldehyde or ketone, and finally to the formation of carboxylic acids [128].

As already seen in PE, the developed upcycling strategies for PP rely on high temperature depolymerization steps (Table 4). Johnston et al. employed thermal oxidation in the presence of oxygen-ozone as a first step to oxidize PP. Then, the thermal-oxidized PP was emulsified in the TSB media before being used as a feedstock for PHA production by *Cupriavidus necator* H16, in shake flasks. After 48 h, 1.36 g/L of PHA was accumulated, accounting for 42% of CDW [129]. Another research group developed a combined process using pyrolysis (540 °C) and yeast fermentation (with *Yarrowia lipolytica*), to upcycle PP into fatty acids (C18 and C16) [130,131].

### 3.4. Bioupcycling of Polystyrene (PS)

PS reached 6.1% of European plastic demand in 2020 [3] and 4.8% worldwide in 2019 [1]. Thanks to its structural stability, it has been used for many purposes ranging from food packaging, household gadgets, electrical devices, and building insulation. The main building block of PS is styrene, which can be combined with additives, colorants, and/or other plastics [15]. As plastic waste has become a pollutant, contaminating landfills and the marine environment, it should be mentioned that one-third of landfilling plastic is PS [132]. Styrene monomers leaching from PS are also very dangerous, being considered carcinogenic in humans [133].

Nonetheless, biodegradation approaches have been studied also for PS. Since it is also a thermoplastic that resists hydrolysis, PS needs a preliminary oxidation process to generate hydrolyzable functional groups. Kim et al. confirmed that carbonyl groups were produced during PS degradation by *Pseudomonas aeruginosa* strain DSM 50071 (isolated from the gut of the superworms *Zophobas atratus*), associated with the enzymes serine hydrolase and S-formylglutathione hydrolase [134]. Ho et al. gathered further information on microbial degradation of PS, identifying relevant fungi, bacteria, and archaea strains [135], and several other studies have isolated PS degrading strains from various sources, such as marine environments [136,137] and gut microbiota of invertebrate larvae [134,138,139].

A previous attempt to turn PS waste into valuable products was performed by Savoldelli and colleagues [132]. They used the combination of thermal (240 °C) and bacterial (*Pseudomonas putida*) degradation to break down PS to naphthalene derivatives, benzene derivatives, as well as some styrene; however, this study did not show the full upcycling of PS. Bioupcycling of PS was, however, successfully performed through another combined approach, using pyrolysis followed by bacterial conversion (Table 5). The fermentation of the styrene pyrolysis oil in a 7.5 L stirred tank reactor accumulated 57% PHA of CDW with a yield of around 0.1 gPHA/g styrene oil (10% conversion). The PHA was characterized as mclPHA with the monomers (R)-3-hydroxyhexanoate, (R)-3-hydroxyoctanoate and (R)-3-hydroxydecanoate [140]. An increase in PHA titer and yield was found when styrene oil after pyrolysis was distilled before being used as a substrate for fermentation [141].

Thermal oxidation with a pro-oxidant/pro-degradant additive of PS coupled with microbial fermentation was also reported to upcycle post-consumer PS to PHA. PS was subjected to thermal oxidation for 20 h at 60 °C (with a flow rate of the oxygen-ozone mixture of 7.5 L/h) and then used as the feedstock for PHA fermentation by *Cupriavidus necator* H16, cultured in tryptone soya broth (TSB). The results showed PHA accumulation at 48% of CDW, accounting for 1.72 g/L. The major subunit was 3-hydroxybutyrate, with up to 12 mol% of 3-hydroxyvalerate and 3- hydroxyhexanoate co-monomeric units [15].

### 3.5. Bioupcycling of Polyvinyl Chloride (PVC)

PVC has low biodegradability, but possesses the highest proportion of plasticizer among other synthetic plastics, which is susceptible to microbial attack [144]. The study by Giacomucci and colleagues investigated the ability of five microbial strains (*Pseudomonas chlororaphis* (DSM 50083), *Pseudomonas citronellolis* (DSM 50332), *Bacillus subtilis* subsp. *spizizenii* (DSM 15029), *Bacillus flexus* (DSM 1320), and *Chelatococcus daeguensis* (DSM 22069)) to biodegrade plastic, showing that a microbial biofilm was found after 3-month incubation of PVC containing 30% (*w*/*w*) of plasticizers, while no biofilm formation was detected on PE, PP, and PS during the same period [145]. *Tenebrio molitor* larvae were reported to reduce weight-, number- and size-average molecular weights (Mw, Mn and Mz) of PVC by 33.4%, 32.8% and 36.4%, respectively, in 16 days. The PVC depolymerization was found to be a gut microbe-dependent biodegradation associated with four families: *Streptococcaceae*, *Spiroplasmataceae*, *Enterobacteriaceae*, and *Clostridiaceae* [146]. Another study investigated the anaerobic biodegradation of PVC by marine consortia and reported gravimetric weight losses of 12% over 7 months [147]. PVC biodegradation by fungi was reported by Ali et al., showing that *Phanerochaete chrysosporium* PV1 caused a significant reduction in the molecular weight of PVC and appearance and shifting of FTIR peaks at the 2370–2350 cm^−1^ region, corresponding to HC-Cl [148].

Also in this case, thermochemical methods proved to be a valuable approach to upcycle PVC into valuable products. For example, a one-pot process including a sequence of dechlorination by Cl-fixative (ZnO or KOH), carbonization of dechlorinated polyenes and further modifications, was used to upcycle PVC into valuable carbon materials, pipeline-quality pyrolysis gas, and chlorides [149]. However, to our knowledge, bioupcycling of PVC, using only bioprocesses, has not been reported so far. The pyrolysis products from PVC, especially those with removed chlorines [150], could be the interesting substrate for further bioconversion steps, such as in the case of recalcitrant polyolefins.

### 3.6. Bioupcycling of Mixed Plastic Waste

The ability to upcycle unsorted, mixed plastic waste would represent a key technology to address real post-consumer plastic streams [151]. To the best of our knowledge, bioupcycling of real mixed plastic waste has hardly been addressed so far. The most promising technology going in this direction was recently proposed by Sullivan and colleagues, that valorized mixed plastic waste (obtained by mixing HDPE, polystyrene and PET), through chemical oxidation coupled to biological funneling [152]. Basically, the polymers were treated with metal-promoted autoxidation that deconstructed them into a mixture of oxygenated intermediates. The latter have an enhanced water solubility, which facilitates their use as a feedstock for the following bioupcycling step, by an engineered *Pseudomonas putida* strain. Besides this study, the selective recovery of monomers in mixed plastic stream has also been proposed, by using enzymes cocktails and microbial mixed consortia processes [151]. The two-enzyme PETase/MHETase system was reported to improve PET and MHET conversion to their monomers, and was highlighted as a model for the future development of multienzyme systems for depolymerization of mixed polymer wastes [60]. Another study by Edwards and team investigated the microbial community-based degradation of diverse carbon sources. The consortium of 2 *Bacillus* and 3 *Pseudomonas* strains was able to degrade PET and also utilize putrescine, an alkane substrate, as a carbon source indicating the potential ability to degrade polyolefins. The strains were also capable of degrading common plasticizers, including phthalates, paraben, and other aromatic and phenolic compounds [153]. This result presents a possibility of using defined mixed consortia for bio-recycling and upcycling of mixed plastic stream.

## 4. Future Perspectives and Conclusions

Plastic is a cheap and ubiquitous material in our society that has been used for multiple functions and applications, to the point to be considered a major characteristic of the current period (the so-called “plastic age”). Even if plastics have countless benefits, they also create a complex waste problem on a global scale. The current linear plastic lifecycle causes environmental pollution, and the negative effects are becoming more and more visible, indicating a non-sustainable value chain. Moreover, the way that plastics are produced, used, and discarded leads to a continuous loss of material from the loop and fails to comply with the circular economy approach. To tackle this problem, the EU has established the waste framework directive, introducing a waste five-step hierarchy that sets the order of preference for managing and disposing of waste (Figure 5) [154]. Preventing unnecessary plastic usage, reducing whenever possible single-use plastics, and reusing plastic materials as much as possible, are the preferred options (higher priority in the hierarchy) to lower amount of plastic waste generated and the carbon footprint. Landfilling is the least desirable, as it causes the loss of material and often leads to the contamination of the environment. The EU commission is thus pushing toward “end-of-waste” or “zero waste” criteria, where new technologies, policies and supply chains will allow us to consider the waste streams as a secondary raw material that should be recovered, reused, or recycled at the highest possible level. Currently, the mechanical recycling is the dominant technology, but, as already mentioned, it does not involve the higher level of the waste hierarchy (reduce and reuse; Figure 5), and it fails to handle contaminated/mixed/multilayered plastics. Moreover, it inevitably leads to downcycling, already after a limited number of cycles. In this sense, bioupcycling can contribute better to the circular economy, by converting post-consumer plastic via (integrated) biotechnological processes to more renewable and carbon-neutral high-value chemicals and new polymers, or at least to recycle the monomers from biodepolymerization to the same type of plastic with preserved quality.

The conversion of fossil-based plastic waste to bioplastics also brings a new opportunity in terms of advanced/add-on properties, such as the improved barrier properties of PEF compared to PET [13]. It also opens the window towards new/innovative materials with different properties from conventional plastics, such as functionalized plastics or the new poly(amide urethane) (bio-PU) obtained from upcycled PET waste (to HAA) [16]. Moreover, the new materials can be developed to be more readily recyclable than the parent polymers, to align with the ambitious goal of Europe’s new plastic strategy that requires 55% of plastic packaging being recyclable by 2030 [4]. A recent study by Roux and Varrone showed that bioupcycling of fossil-derived PET to bio-PTT and bio-PEF can represent an important drive towards plastic waste valorization and increased recycling rates [12]. In fact, the techno-economic analysis showed that recycling PEF and PTT to the required 55% would decrease their production cost by 50%, obtaining a minimum selling price of 1.61 USD/Kg, which is lower than rPET prices [47]. In this consideration, bioupcycling is expected to play an important role towards a smarter, more innovative, and more sustainable plastic sector.

Due to the incredible complexity and diversity of plastic waste, the future success of upcycling approaches will ultimately depend on the proper combination and integration of different complementary technologies and processes. In this sense, cross-disciplinarity will be a key enabler. A good example is the sequential process of chemical depolymerization of PET, followed by biological conversion of PET hydrolysates to higher values building blocks. Chemocatalytic glycolysis produces, for instance, BHET, which can be bioconverted by engineered *P. putida* to β-ketoadipic acid, a valuable monomer for the production of a performance-advantaged nylon-6,6 analog polymer [73]. Other chemical depolymerization strategies (e.g., methanolysis to produce dimethyl terephthalate, aminolysis to produce terephthalamide, or alcoholysis to produce terephthalate diesters [155,156]) could be combined with biotechnological upcycling as well. The incredible diversity of metabolic pathways and enzymes can be exploited to develop new bioprocesses to valorize the complex depolymerization broths obtained from thermochemical processes. New high-value products can be obtained from specific hydrolysates (without previous purification), by the use of consolidated bioprocesses and synthetic biology. Moreover, while thermochemical processes can boost the (bio)degradation and upcycling steps, biological processes (such as enzyme technology) might remove specific unwanted compounds and impurities to increase overall thermochemical conversion yields (i.e., removing PA before pyrolysis to avoid formation of cyanide gasses and costly downstream gas cleaning), thus overcoming mutual limitations.

According to the plastics microbial biodegradation database (PMBD), 949 microorganism–plastic relationships and 79 genes involved in the biodegradation of plastics have been reported [157]. There is no doubt that these numbers are going to dramatically increase in the coming years thanks to advanced metagenomic and proteomic approaches, new modeling tools, new synthetic biology, and protein engineering techniques [158]. The development of new high-throughput screening methods will be paramount. The constant discovery and engineering of microbes and enzymes that can selectively degrade certain types of polymers will enable more effective recycling of dirty mixed plastic waste fractions that currently are incinerated or go to landfill. An excellent example is the interesting coupled bioleaching/enzyme-based process for the recycling of multilayer packaging that was recently developed by Kremser and colleagues [159]. The study successfully demonstrated an innovative method to recover pure PE and aluminum hydroxide from beverage cartons. The achievement of this study opens for the biological recycling of multilayer materials. All these findings anticipate the possibility of developing novel biocatalysts and bioprocesses that will allow for cost-effective and scalable plastic bioupcycling.

In conclusion, it is important to underline that only an extra effort on assessing economic feasibility and environmental impacts of such coupled biological processes will illustrate their viability and contribution to a circular economy.

## Figures and Tables

**Figure 1 polymers-14-04996-f001:**
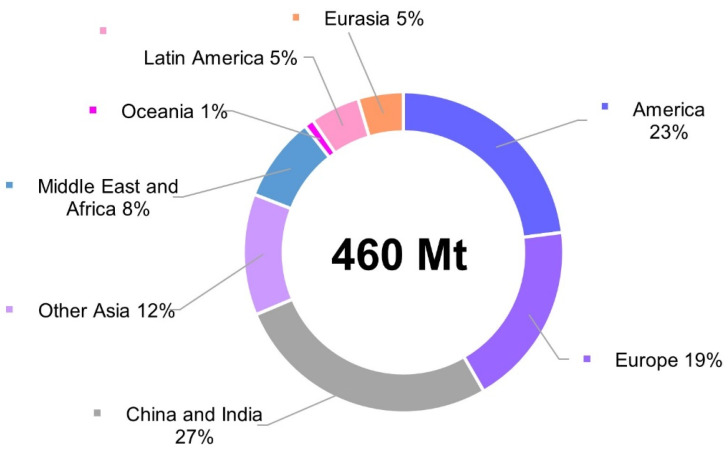
Global plastic use by region. The figure is created according to the data from OECD [1].

**Figure 2 polymers-14-04996-f002:**
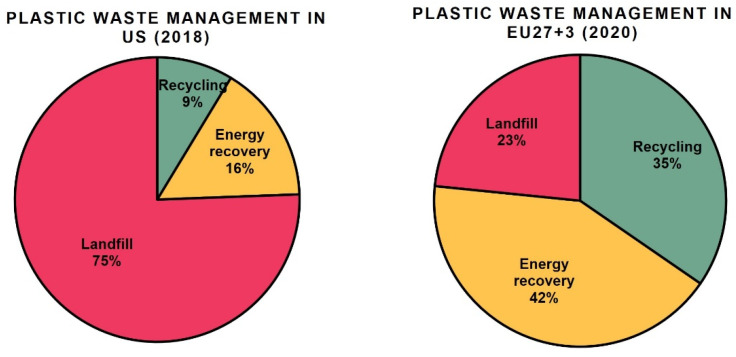
Post-consumer plastic waste management in US (2018) and EU27+3 (2020). The figure is created according to the data from US EPA [2] and PlasticsEurope [3].

**Figure 3 polymers-14-04996-f003:**
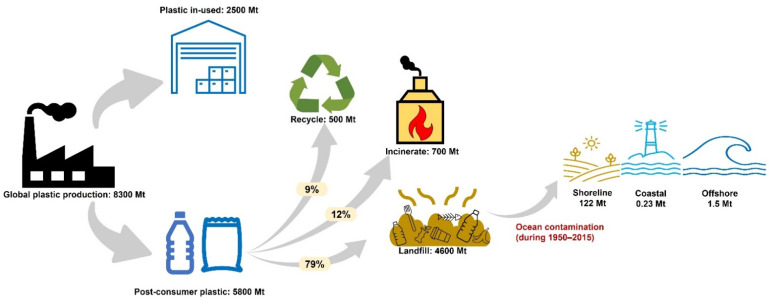
The pathway of plastics from primary production until end-of-life, during the period 1950–2015, based on the study of Geyer et al. (2017) [9] and Lebreton et al. (2019) [10].

**Figure 4 polymers-14-04996-f004:**
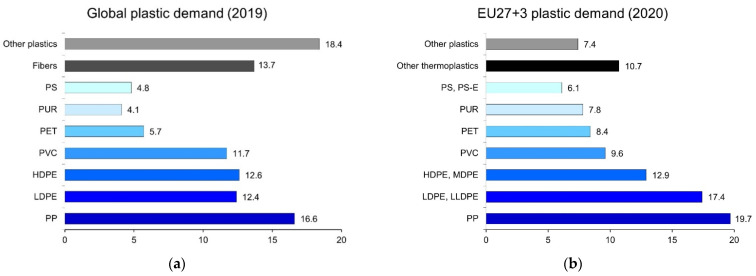
Distribution of plastic demand globally (**a**) and in EU27+3 (**b**) by polymer types. The figure is created according to the data from OECD [1] and PlasticsEurope [3].

**Figure 5 polymers-14-04996-f005:**
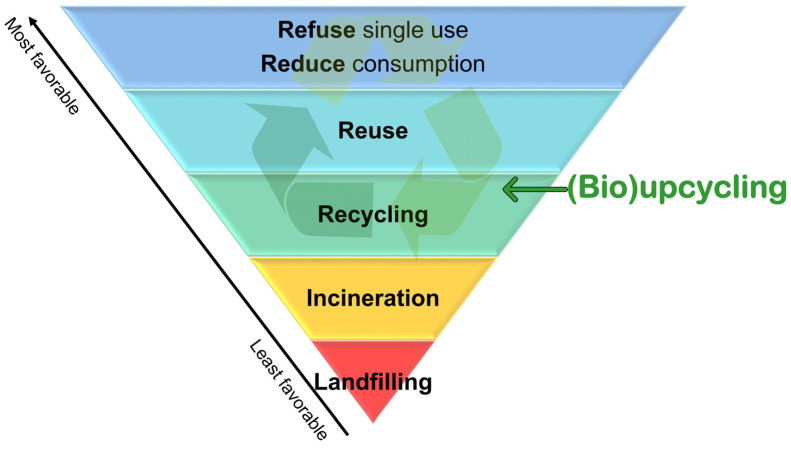
The hierarchy of plastic waste management according to EU Waste Framework Directive.

**Table 2 polymers-14-04996-t002:** PU bioupcycling.

Depolymerization Strategy	Depolymerization Products Used as a Feedstock for Fermentation Step	Fermentation Strategy	Products from Fermentation	Titer	Productivity	Yield	Ref.
Enzymatic degradation of polycaprolactone polyol-based PU by esterase (E3576) in 0.1 M phosphate buffer (pH 7.0). The enzyme solution was replaced every 3–4 d to overcome a loss of enzymatic activity.	6-hydroxycaproic acid (1 g/L)	-	-	-	-	-	[93]
-	Adipic acid (AA) (mock substrate to study upcycling of PU-derived monomer)	Bioconversion (at 30 °C and 200 rpm for 135 h) using metabolically engineered *P. putida* KT2440 A12.1p pPS05 to convert AA into HAA and then to rhamnolipid	Rhamnolipid	0.02 g/L	NA	0.014 gRhamnolipid/gSubstrate	[75]
-	1,4-Butanediol (BDO) (mock substrate to study upcycling of PU-derived monomer)	Bioconversion (at 30 °C and 200 rpm for 135 h) using metabolically engineered *P. putida* KT2440 B10.1 pPR05 to convert BDO into HAA and then to rhamnolipid	Rhamnolipid	0.13 g/L	NA	0.088 gRhamnolipid/gSubstrate	[75]
-	EG (mock substrate to study upcycling of PU-derived monomer)	Bioconversion (at 30 °C and 200 rpm for 135 h) using metabolically engineered *P. putida* KT2440 ∆*gclR* ∆PP_2046 ∆PP_2662::14d to convert EG into HAA and then to rhamnolipid	Rhamnolipid	0.07 g/L	NA	0.038 gRhamnolipid/gSubstrate	[75]
-	AA + BDO + EG (mock hydrolysate to study upcycling of PU-derived monomers)	Bioconversion (at 30 °C and 200 rpm for 210 h) using mixed culture of three metabolically engineered *P. putida* KT2440 to convert the mock hydrolysate into HAA and then to rhamnolipid	Rhamnolipid	0.1 g/L	NA	0.008 gRhamnolipid/gSubstrate	[75]
	AA + BDO + EG + 2,4-toluenediamine (TDA) (mock hydrolysate to study upcycling of PU-derived monomers)	Bioconversion (at 30 °C and 200 rpm for 210 h) using mixed culture of three metabolically engineered *P. putida* KT2440 to convert the mock hydrolysate into HAA and then to rhamnolipid without extraction of TDA	Rhamnolipid	0.02 g/L	NA	0.002 gRhamnolipid/gSubstrate	[75]
	AA + BDO + EG + TDA (mock hydrolysate to study upcycling of PU-derived monomers)	Bioconversion (at 30 °C and 200 rpm for 210 h) using mixed culture of three metabolically engineered *P. putida* KT2440 to convert the mock hydrolysate into HAA and then to rhamnolipid with extraction of TDA at pH 3.5	Rhamnolipid	0.07 g/L	NA	0.005 gRhamnolipid/gSubstrate	[75]

**Table 4 polymers-14-04996-t004:** PP bioupcycling.

Depolymerization Strategy	Depolymerization Products Used as a Feedstock for Fermentation Step	Fermentation Strategy	Products from Fermentation	Titer	Productivity	Yield	Ref.
Pro-degradation at 180 °C with 1% (*w*/*w*) cobalt stearate as pro-oxidant/pro-degradant additive	Oxidatively pro-degraded PP	Fermentation in shake flask containing 2 g/L oxidatively pro-degraded PP emulsified in TSB by sonication as a sole carbon source at 30 °C for 48 h by *C. necator* H16	PHA	0.58 g/L	NA	0.26 gPHA/gCDW	[129]
Oxidatively pro-degraded PP was subjected to oxidative degradation in a two-phase system (gas-liquid phase), after melting at 60–80 °C and using oxygen-ozone mixture	Thermal oxidized PP	Fermentation in shake flask containing 2 g/L thermal-oxidized PP emulsified in TSB by sonication as a sole carbon source at 30 °C for 48 h by *C. necator* H16	PHA	1.36 g/L	NA	0.42 gPHA/gCDW	[129]
Pyrolysis at 540 °C	Pyrolysis oil contained branched chain fatty alcohols (51%) and alkenes (25%)	Fermentation in shake flask containing OP4 medium (15 g/L pyrolysis oil, 5.4 g/L Tween 80, 4.5 g/L oleic acid, 1.25 g/L (NH_4_)_2_SO_4_, 2.5 g/L KH_2_PO_4_, and 0.830 g/L MgSO_4_·7H_2_O) at 30 °C for 312 h by *Yarrowia lipolytica* strain 78-003	Fatty acids with C18 compounds (oleic acid, linoleic acid, and stearic acid) as dominant products, followed by C16 compounds (palmitic and palmitoleic acids).	492 mg/L	NA	NA	[130]

**Table 5 polymers-14-04996-t005:** PS bioupcycling.

Depolymerization Strategy	Depolymerization Products Used as a Feedstock for Fermentation Step	Fermentation Strategy	Products from Fermentation	Titer	Productivity	Yield	Ref.
-	Styrene (mock substrate to study upcycling of PS-derived monomer)	Fermentation in shake flask containing 1.85 g/L styrene as a sole carbon source and 67 gN/L NaNH_4_HPO_4_·4H_2_O as a nitrogen source at 30 °C for 48 h by *P. putida* CA-3	PHA	NA	NA	0.099 gPHA/gStyrene	[142]
Pyrolysis at 520 °C	Styrene oil (82.8% (*w*/*w*) styrene as well as low level of α-methylstyrene, toluene, styrene dimer, and traces of other aromatic compounds)	Fermentation in shake flask containing styrene oil as a sole carbon source and 1 g/L NaNH_4_HPO_4_·4H_2_O as a nitrogen source at 30 °C by *P. putida* CA-3	PHA	0.14 g/L	NA	0.0625 gPHA/gStyrene oil (0.25 gPHA/gCDW)	[140]
Pyrolysis at 520 °C	Styrene oil (82.8% (*w*/*w*) styrene as well as low level of α-methylstyrene, toluene, styrene dimer, and traces of other aromatic compounds)	Fermentation in 7.5 L stirred tank reactor feeding a sole carbon source through the gaseous phase contained styrene oil at a concentration of 9.5 mg/L (flow rate 0.15 L/min for the first 3 h of growth and increased to 0.25 L/min for the subsequent 3 h, and finally, to 0.65 L/min for the remainder) at 30 °C by *P. putida* CA-3	PHA	0.32 g/L	NA	0.1 gPHA/gStyrene oil (0.57 gPHA/gCDW)	[140]
Pyrolysis	Distilled styrene oil (89.9% styrene, 5.63% α-methylbenzene, 2.63% toluene, 1.05% ehtylbenzene, 0.43% benzene, 0.19% 1-ethyl-2-methy benzene, and 0.17% unknown)	Fermentation in stirred tank reactor feeding distilled styrene oil at a feed rate of 75 mg/L/h (equivalent to 69 mgC/L/h) and NaNH_4_HPO_4_·4H_2_O at a feed rate of 1.5 mg/L/h at 30 °C by *P. putida* CA-3	PHA	0.82 g/L	NA	0.28 gPHA/gStyrene oil (0.42 gPHA/gCDW_)_	[141]
-	Styrene (mock substrate to study upcycling of PS-derived monomer)	Fed-batch fermentation in stirred tank reactor feeding styrene as a carbon source overtime through air sparger and NH_4_Cl as a nitrogen source at different feed rate during the operation period. The fermentation was conducted at 30 °C and pH 6.9 by *P. putida* CA-3	mclPHA	3.36 g/L	NA	0.32 gPHA/gCDW	[143]
Pro-degradation	Pro-degraded PS	Fermentation in shake flask containing 3.7 g/L prodegraded PS emulsified in TSB by sonication as a sole carbon source at 30 °C for 48 h by *C. necator* H16	PHA	0.52 g/L	NA	0.39 gPHA/gCDW	[15]
Pro-degraded PP was subjected to thermal oxidation (60 °C) in a two-phase system (gas-solid phase) using oxygen-ozone mixture	Thermal oxidized PS (60 °C)	Fermentation in shake flask containing 3.7 g/L thermal oxidized PS emulsified in TSB by sonication as a sole carbon source at 30 °C for 48 h by *C. necator* H17	PHA	1.72 g/L	NA	0.48 gPHA/gCDW	[15]
Pro-degraded PP was subjected to thermal oxidation (80 °C) in a two-phase system (gas-solid phase) using oxygen-ozone mixture	Thermal oxidized PS (80 °C)	Fermentation in shake flask containing 3.7 g/L thermal oxidized PS emulsified in TSB by sonication as a sole carbon source at 30 °C for 48 h by *C. necator* H18	PHA	1.28 g/L	NA	0.42 gPHA/gCDW	[15]
Pro-degraded PP was subjected to thermal oxidation (100 °C) in a two-phase system (gas-solid phase) using oxygen-ozone mixture	Thermal oxidized PS (100 °C)	Fermentation in shake flask containing 3.7 g/L thermal oxidized PS emulsified in TSB by sonication as a sole carbon source at 30 °C for 48 h by *C. necator* H19	PHA	0.96 g/L	NA	0.36 gPHA/gCDW	[15]

## Data Availability

Not applicable.

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
