# Peer review of "Critical Review on the Progress of Plastic Bioupcycling Technology as a Potential Solution for Sustainable Plastic Waste Management"

_polymers, 2022, doi:10.3390/polym14224996_

Round 1

Reviewer 1 Report

Very well done! One of the easiest peer-reviews I've ever had to do. 

This comprehensive literature review on bio-upcycling covered the up-and-coming technologies related to chemo-biological recycling of industrial plastics and other polymers. The authors described the basics of mechanical, chemical, and biological recycling methods and then dove deeper into the use of biological mechanisms to degrade specific polymer types into molecules chemically similar or dissimilar to their original polymer feedstocks. 

The article is easily understood for a wide audience of readers. The authors did a good job of writing in plain language (when possible), and the grammar is well edited. There are only minor suggestions within the uploaded pdf comments. Also, the well-organized tables of bio-upcycling techniques per polymer were very much appreciated in such a lengthy, well-supported review article. 

I fully support the acceptance and publishing of this article, well done. 

Author Response

Thank you very much for the encouraging words. We have addressed the reviewer’s comments and corrected the identified mistakes in Line 247 and 479. Moreover, we have added references to support our statement in Line 374-375.

Reviewer 2 Report

The authors presented a comprehensive review on the plastic bioupcycling technology, which can give instructive guidance for the future study about sustainable plastic waste upgrading. Detailed and extensive researches featuring the bioupcycling methods, mechanism and outcomes have been reviewed by referring most of the related studies, and critical perspectives were derived for the future attentions. Minor revision is needed as following:

(1)   The published reviews about the plastic waste management should be addressed to highlight the novelty of this review.

(2)   In the table referring the bioupcycling of different types of plastic materials, the corresponding setup for the process is suggested to be mentioned.

(3)   The authors have reviewed the upgrading of single type of plastic, how about the technology for complex plastic.

Author Response

1) The published reviews about the plastic waste management should be addressed to highlight the novelty of this review.

We have now mentioned the published reviews about plastic waste management and discussed briefly about what is still missing in those publications to highlight the important of this review, Lines 105-109.

“Most of the reviews dealing with plastic waste management focus on conventional processes, e.g., landfilling, incineration, mechanical, and chemical recycling [19–21] that downcycle the materials. They do not analyze the advantages of integrated processes that combine thermochemical and biochemical technologies..”

2) In the table referring the bioupcycling of different types of plastic materials, the corresponding setup for the process is suggested to be mentioned.

Additional information about the process condition have been added in Table 1 (PET bioupcycling) and Table 2 (PU bioupcycling), including processes’ temperature, mixing speed, time, and media details (e.g., composition and pH), as reviewer suggested.

(3)   The authors have reviewed the upgrading of single type of plastic, how about the technology for complex plastic.

We would like to thank the reviewer to remarking this important point. The upcycling of mixed plastic has been emphasized as an ultimate goal in several studies, as it represents the real waste stream. However, to the best of our knowledge, there is no publication reporting about the successful strategy on real post-consumer mixed plastic waste yet. Nevertheless, the concept has been explained in several publications and there are several EU projects looking into it (mentioned at Lines 84-88) and an interesting study on mixed plastic upcycling was just published in Science, 3 weeks ago. We agree that it is important to discuss more about the mixed plastic, thus we have added a new section, discussing about current state of bioupcycling of mixed plastic waste (Lines 650-671)

“3.6. Bioupcycling of mixed plastic waste

The ability to upcycle unsorted, mixed plastic waste would represent a key technol-ogy to address real post-consumer plastic streams [151]. To the best of our knowledge, bi-oupcycling of real mixed plastic waste has hardly been addresses so far. The most prom-ising technology going in this direction was recently proposed by Sullivan and colleagues, that valorized mixed plastic waste (obtained by mixing HDPE, polystyrene and PET), through chemical oxidation coupled to biological funneling [152]. Basically, the polymers were treated with metal-promoted autoxidation that deconstructed them into a mixture of oxygenated intermediates. The latter have an enhanced water solubility, which facilitates their use as a feedstock for the following bioupcycling step, by an engineered Pseudomonas putida strain. Besides this study, the selective recovery of monomers in mixed plastic stream has also been proposed, by using enzymes cocktails and microbial mixed consor-tia processes [151]. The two-enzyme PETase/MHETase system was reported to improve PET and MHET conversion to their monomers, and was highlighted as a model for future development of multienzyme systems for depolymerization of mixed polymer wastes [60]. Another study by Edwards and team investigated microbial community-based degrada-tion of diverse carbon sources. The consortium of 2 Bacillus and 3 Pseudomonas strains was able to degrade PET and also utilize putrescine, an alkane substrate, as a carbon source indicating potential ability to degrade polyolefins. The strains were also capable of de-grade common plasticizers, including phthalates, paraben, and other aromatic and phe-nolic compounds [153]. This result presents a possibility of using defined mixed consortia for bio-recycling and upcycling of mixed plastic stream.”

Moreover, the conclusion about (bio)recycling of mixed and multilayer plastic waste has already mentioned at Lines 735-743, with an interesting example (bioleaching/enzyme-based process for the recycling of multilayer packaging).
